# *Salmonella typhimurium* Vaccine Candidate Delivering Infectious Bronchitis Virus S1 Protein to Induce Protection

**DOI:** 10.3390/biom14010133

**Published:** 2024-01-20

**Authors:** Kaihui Liu, Zewei Li, Quan Li, Shifeng Wang, Roy Curtiss, Huoying Shi

**Affiliations:** 1College of Veterinary Medicine, Yangzhou University, Yangzhou 225009, China; mx120190697@stu.yzu.edu.cn (K.L.); dz120190005@stu.yzu.edu.cn (Z.L.); liquan2018@yzu.edu.cn (Q.L.); 2Jiangsu Co-Innovation Center for the Prevention and Control of Important Animal Infectious Diseases and Zoonoses, Yangzhou 225009, China; 3Department of Infectious Diseases and Immunology, College of Veterinary Medicine, University of Florida, Gainesville, FL 32611, USA; shifengwang@ufl.edu (S.W.);; 4Joint International Research Laboratory of Agriculture and Agri-Product Safety, Yangzhou University (JIRLAAPS), Yangzhou 225009, China

**Keywords:** avian infectious bronchitis virus, S1 protein, regulated delayed attenuation and lysis system, *S. typhimurium*, immune protection

## Abstract

Infectious bronchitis (IB) is a highly infectious viral disease of chickens which causes significant economic losses in the poultry industry worldwide. An effective vaccine against IB is urgently needed to provide both biosafety and high-efficiency immune protection. In this study, the S1 protein of the infectious bronchitis virus was delivered by a recombinant attenuated *Salmonella typhimurium* vector to form the vaccine candidate χ11246(pYA4545-S1). *S. typhimurium* χ11246 carried a *sifA*^−^ mutation with regulated delayed systems, striking a balance between host safety and immunogenicity. Here, we demonstrated that S1 protein is highly expressed in HD11 cells. Immunization with χ11246(pYA4545-S1) induced the production of antibody and cytokine, leading to an effective immune response against IB. Oral immunization with χ11246(pYA4545-S1) provided 72%, 56%, and 56% protection in the lacrimal gland, trachea, and cloaca against infectious bronchitis virus infection, respectively. Furthermore, it significantly reduced histopathological lesions in chickens. Together, this study provides a new idea for the prevention of IB.

## 1. Introduction

Avian infectious bronchitis (IB) is an acute and highly infectious respiratory disease of chickens caused by the infectious bronchitis virus (IBV). It mainly affects the respiratory tract, kidney, and reproductive system in chickens; some IBV variants also affect the intestinal tract and bursa of Fabricius [1,2,3]. IBV belongs to the γ coronavirus and has a single-stranded, positive-sense RNA genome of 27.6 kp that encodes four structural proteins: spike protein (S), membrane protein (M), envelope protein (E), and nucleocapsid protein (N) [4]. The S protein can be hydrolyzed by protease into two subunits, S1 and S2. The S1 subunit is responsible for attaching the viruses to cells and inducing the production of neutralizing antibodies to provide protective immunity [5,6]. Therefore, it is a major target for the design of new IBV vaccines.

The continuous emergence of new types and variants of IBV, driven by rapid mutation rates, virus recombination, and host selection pressures, poses a global challenge for IB prevention [7,8]. Currently, vaccination remains the most effective means to prevent and control IB in the poultry industry [9]. The widely used commercial IBV vaccines on the market are live attenuated and inactivated vaccines [10]. Live attenuated IBV vaccines are effective; however, they carry the risk of reversion to virulence and often involve co-infection with other avian viruses, such as the H9N2 avian influenza virus, resulting in more severe outbreaks [11]. Inactivated IBVs are safe but costly, and elevated levels of protection will be achieved by co-administration with live attenuated vaccines [12,13]. Therefore, there is a need to develop a new vaccine, which not only does not depend on chicken embryos but is also cost-effective and capable of inducing high-efficiency and long-lasting immune protection [14]. An alternative approach could be DNA vaccines, which offer durable expression cycle and good cellular responses with improved safety [15,16]. However, DNA vaccines have been associated with weak and slow humoral responses, and their method of inoculation is not suitable for large-scale commercial chicken farming [17]. Thus, an efficient delivery system is required to overcome these challenges and enhance the effectiveness of DNA vaccines.

Recombinant attenuated *S. typhimurium* vaccine (RASV) has been used as an effective DNA vaccine delivery system for inducing mucosal, humoral, and cellular immunity against heterologous antigens through oral immunization [18,19,20]. Oral RASV initially colonizes in the gut-associated lymphoid tissue via M cells of Peyer’s patches or is taken up by intestinal epithelial goblet cells, subsequently colonizing in deep lymphoid tissues such as the liver and spleen [21]. However, RASV resides in the *Salmonella*-containing vacuole (SCV) within the host cytoplasm to evade clearance by the host immune system. The SCV inhibits antigen presentation and immune responses, leading to reduced protection efficiency against viral infection [22]. A novel regulated delayed attenuation and lysis *S. typhimurium* χ11246, developed by the research team of Dr. Roy Curtiss III [23], addresses this concern. The SCV formation by *Salmonella* is prevented by deleting the *sifA* gene (*sifA^−^*), which is involved in the biogenesis of the *Salmonella* SCV. As a result, *Salmonella* within the SCV can be released into the host cytoplasm. In addition, a regulated delayed lysis in vivo system based on a Δ*asdA* mutation and insertion of an arabinose-regulated expression of the chromosomal *murA* gene, along with a plasmid vector-encoding arabinose-regulatable expression of *asdA* and *murA* genes, conferred attenuation and biological containment [22]. The *Salmonella* lyses and releases the plasmid-carrying heterologous virus DNA into the host cytoplasm when invading an arabinose-free host environment. Subsequently, the plasmid will be ingested by the nucleus, and heterologous virus DNA will be expressed to induce humoral and cellular immune responses [24,25,26]. The system has been shown to improve the efficiency of immune protection provided by the delivery of influenza virus HA and NP proteins [22]. Furthermore, oral RASV has been demonstrated to induce a robust mucosal immune response [27].

In this study, a regulated delayed attenuation and lysis *S. typhimurium* vaccine candidate χ11246(pYA4545-S1) was constructed to deliver the heterologous protective antigen S1 from serotype 793/B IBV. The immunogenicity and the protection efficiency of χ11246(pYA4545-S1) against IBV were evaluated in SPF chicken.

## 2. Materials and Methods

### 2.1. Ethical Statement

All procedures involving the care and use of animals were conducted to minimize animal suffering and approved by the Jiangsu Laboratory Animal Welfare and Ethics guidelines of the Jiangsu Administrative Committee for Laboratory Animals (SCXK2018-0009 and SYXK 2021-0027 for chickens and SCXK2017-0007 and SYXK2017-0044 for mice). The mice and chickens used in this study were both euthanized by manual cervical dislocation [13,28]. The SPF broilers and chicken embryos used in this study were purchased from Nanjing Biology Medical Factory, Qian Yuan-hao Biological Co., Ltd., Nanjing, China. Female 6-week-old BALB/c mice were obtained from the Comparative Medicine Center of Yangzhou University.

### 2.2. Bacterial Strains, Cells, and Plasmids

The strains and plasmids utilized in this study are listed in Table 1. *S. typhimurium* strain χ11246 is derived from the highly virulent strain UK-1 wild-type χ3761 [29]. *E. coli* strains DH5α and BL21 (DE3) were purchased from Invitrogen for expression of the recombinant plasmids pET28a-S1. *S. typhimurium* strain χ11246 (*sifA*^−^), *E. coli* strain χ7213, and plasmid pYA4545 were kindly provided by Dr. Roy Curtiss III [22,23]. HD11 cells were cultured at 37 °C with 5% CO_2_ in Dulbecco’s modified Eagle’s medium (DMEM, C11960500BT, Gibco, Waltham, MA, USA), supplemented with 10% fetal bovine serum (FBS), 100 U/mL penicillin, and 100 μg/mL streptomycin at pH 7.2. *E. coli*, and *S. typhimurium* strains were grown in LB broth or on LB [30] agar plates at 37 °C. Kanamycin (Kan: 50 mg/mL) and ampicillin (Amp: 100 mg/mL) were added for the construction of plasmid pET28a-S1 and plasmid pMD19T-S1, respectively. The growth of Asd^−^ strains depended on the supplement of DAP at a concentration of 50 μg/mL [31]. For host-regulated delayed lysis vector combinations, 0.2% arabinose and 0.2% mannose were used in LB.

### 2.3. Virus Strain Propagation and Purification

Serotype 793/B IBV was isolated and preserved in our laboratory, the virus was serially passaged three times in 10-day-old specific-pathogen-free (SPF) chicken embryos, and the allantoic fluid of the chicken embryos was harvested 36 h after inoculation and stored at −80 °C.

Purified viruses for use in cytokine detection were prepared using differential centrifugation and sucrose banding of virus stocks. The detailed operation was as follows. Briefly, the allantoic fluid was centrifuged at 6000 rpm for 20 min and 8000 rpm for 30 min to obtain the supernatant; the supernatant was continually centrifuged by ultracentrifugation at 30,000 rpm for 3 h, washed in PBS, and the precipitation was suspended with PBS. A volume of 5 mL of 20% sucrose solution was prepared, and the suspension was slowly added to the sucrose solution to show obvious stratification. After being centrifuged at 30,000 rpm for 3 h, the precipitation was resuspended in PBS to obtain purified IBV proteins and stored at −80 °C.

### 2.4. Expression and Purification of Recombinant S1 Proteins and Preparation of Antiserum

Total RNA was extracted from the allantoic fluid harvested above by the RNA-easy Isolation Reagent (R70101, Vazyme, Nanjing, China) and reverse transcribed to cDNA using the Reverse Transcription Master Mix (RR036A, Takara, Dalian, China). The epitope-rich region of the *s1* gene was predicted using DNAstar 11.2.1 software (DNAstar Inc., Madison, WI) and amplified using the primers S1-F1 and S1-R1 (Table 2). The *s1* gene was then inserted into the pET28a vector using EcoRI and XhoI restriction enzymes. The pET28a-S1 plasmid was transformed into *E. coli* strain BL21 (DE3) for expression to obtain S1 protein. *E. coli* strain BL21 carrying pET28a-S1 was grown to mid-log phase (OD600 values of 0.4–0.6) in LB medium with kanamycin at 37 °C and induced with 1 mM IPTG for 4 h. The protein was predominantly found as the inclusion body and was subsequently denatured and renatured using the following methods: the inclusion body was washed twice using PBS and suspended in a washing solution (50 Mm Tris-HCl, 10 mmol/L EDTA, 100 mmol/L NaCl, 1% Triton-X100). It was then centrifuged at 7000 rpm for 15 min at 4 °C, and the collected supernatant was placed in dialysis bags. The bags were sequentially placed in gradient urea renaturation buffer (6 M, 4 M, 2 M, 1 M, 0.5 M) and PBS for renaturation. Finally, the supernatant was collected by centrifugation at 7000 rpm for 10 min at 4 °C. The protein concentration was determined using the BCA protein assay kit (P0012, Beyotime, Shanghai, China). The samples were stored at −80 °C.

Three female 6-week-old BALB/c mice were purchased from the Comparative Medicine Center of Yangzhou University. Following the instructions, a mixture of 500 μg of S1 protein and adjuvant (KX0210042, Biodragon, Beijing, China) was intramuscularly injected into the mice twice with an interval of three weeks between injections. Serum samples were collected two weeks after the second immunization. A titer of ≥1000 was considered positive for the ELISA.

### 2.5. Vectors for Antigen Delivery via Regulated Delayed Attenuation and Lysis System

The coding sequence of the IBV *s1* gene was amplified through gene splicing by overlap extension PCR with the primers S1-P1, S1-P2, S1-P3, and S1-P4, as described in Table 2. The PCR products were cloned into pMD19T vector (Takara, Dalian, China) to generate the plasmid pMD19T-S1. The plasmid pYA4545-S1 was generated by ligating the fragments from plasmids pYA4545 and pMD19T-S1, digested with KpnI and XhoI. Plasmid pYA4545-S1 was confirmed through KpnI/XhoI digestion and gene sequencing. The mutation Δ*sifA* was confirmed in the regulated delayed attenuation and lysis *S. typhimurium* strain χ11246 by PCR. The primers used are given in Table 2. The plasmids pYA4545-S1 and pYA4545 were transformed into strain χ11246 to generate χ11246(pYA4545-S1) and χ11246(pYA4545), respectively.

### 2.6. Analysis of Recombinant Protein Expressed In Vitro

The expression of the recombinant protein S1 in HD11 cells was detected by indirect immunofluorescence assay (IFA) and Western blot. For IFA, HD11 cells were transfected with the 1 μg of recombinant plasmid pYA4545-S1 and the control plasmid pYA4545, respectively, using FuGENE^®^ HD Transfection Reagent (E2311, Promega, Madison, WI, USA) at a ratio of 1:3 (DNA:transfection reagent). The cells were then fixed with cell fixator (acetone:ethanol = 2:3) for 10 min at 48 h post-transfection. The anti-S1 antiserum prepared in this study was used as the primary antibody, while fluorescein isothiocyanate (FITC)-labeled goat anti-mouse IgG antibody (1:5000, bs0296G, Bioss, Beijing, China) was used as the secondary antibody. Fluorescent signals of S1 in HD11 cells were recorded using fluorescent microscopy. For Western blot, HD11 cell lysates transfected with the recombinant plasmid pYA4545-S1 and the control plasmid pYA4545, collected at 48 h post-transfection, were subject to 12% SDS-PAGE. The proteins were transferred to polyvinylidene fluoride (PVDF) membranes and blocked in 5% skim milk with phosphate buffer solution with Tween-20 (PBST) buffer. The antiserum was used as the primary antibody, followed by the horseradish peroxidase HRP-labeled goat anti-mouse antibody IgG (1:5000 dilutions, BA1050, Boster, Wuhan, China). Signals were visualized using the DAB Horseradish Peroxidase Color Development Kit (P0202, Beyotime, Shanghai, China) and scanned with Tanon-5200 Electrophoresis Gel Imaging Analysis System.

### 2.7. Animal Experimental Design and Sample Collection

To evaluate the immunogenicity and protective immunity elicited against IBV by the recombinant *S. typhimurium* strain χ11246(pYA4545-S1), a total of twenty-four 1-day-old chickens were randomly divided into four groups with six chickens in each group. Two groups of chickens were immunized with χ11246(pYA4545-S1) or χ11246(pYA4545) on the day of hatching, at 2 weeks, and at 4 weeks of age, respectively. Another two non-immunized groups were set as the virus-challenged control group and the blank control group. Blood samples were collected at 4 weeks and 6 weeks after the first immunization (i.e., 2 weeks after the second and the third immunization, respectively) and subsequently tested for the presence of antibodies against S1 by ELISA. Peripheral blood mononuclear cells (PBMCs) were isolated and assayed for cytokine secretion after antigenic stimulation at 2 weeks after the third immunization. Chickens in the χ11246(pYA4545-S1) immunized group, χ11246(pYA4545) immunized group, and virus-challenged control group were challenged with serotype 793/B IBV at 2 weeks after the third immunization, while the blank control group was not challenged. Lacrimal gland, oropharynx, and cloacal cotton swabs of each group were collected and placed into 900 μL of PBS with 1% penicillin/streptomycin/amphotericin B solution (C810KA3468, Sangon Biotech, Shanghai, China) three days after the challenge for subsequent viral shedding rate detection. Five days after the challenge, all chickens in each group were euthanized to obtain tissue samples (i.e., lungs, trachea, and kidneys) for histological analysis.

### 2.8. Immunization of SPF Chicken

The recombinant *S. typhimurium* strains χ11246(pYA4545-S1) and χ11246(pYA4545) were each grown to an OD600 of 0.85 in LB broth containing 0.2% arabinose and 0.2% mannose. A final concentration of 5 × 10^9^ CFU/mL was achieved by centrifuging at 4000 rpm for 15 min at room temperature and suspending the cultures in phosphate-buffered saline (PBS). The bacteria were titrated on LB agar with arabinose. After a 4 h fast, each chicken in the χ11246(pYA4545-S1) and χ11246(pYA4545) groups was orally immunized with 200 µL (1 × 10^9^ CFU). Food and water were returned 30 min after vaccine administration.

### 2.9. EID_50_ of IBV Test and Challenge

The 10-day-old SPF chicken embryos were infected with a continuous 10-fold dilution ranging from 10^−3^ to 10^−16^ of the IBV. Each dilution was inoculated into six eggs, and allantoic fluid was collected 36 h later and treated with 1% trypsinase at 37 °C for 3 h for the hemagglutination (HA) test. The 50% egg infectious dose (EID_50_) of serotype 793/B IBV was determined by the Reed–Muench method (Reed and Muench 1938). Chickens were intranasally infected with 10^4^ EID_50_ of serotype 793/B IBV in a total volume of 200 µL.

### 2.10. Enzyme-Linked Immunosorbent Assay (ELISA) for Antibodies

The levels of serum anti-S1 antibodies were measured by ELISA with a homemade ELISA kit. ELISA plates were coated overnight at 4 °C with purified viral protein at a protein concentration of 1 μg/well in 100 μL of coating buffer (bicarbonate/phosphate buffer). After washing with 1 × PBST three times, the plates were blocked for 4 h at room temperature with 5% skim milk. The serum was serially diluted in the blocking buffer to 1:400 and added to the wells (100 μL/well), after which the plates were incubated for 2 h at room temperature. Following three washes with PBST, the plates were incubated with 100 μL/well of HRP-labeled goat anti-mouse antibody IgG (BA1050, Boster, Wuhan, China), diluted 1:5000 in blocking buffer, for 1 h at room temperature. The plates were washed with PBST three times again and then incubated for 10 min at room temperature in the dark with 100 μL of tetramethyl benzidine (TMB) substrate per well (P0209, Solarbio, Beijing, China). The reactions were stopped by adding 50 μL of 0.05 M H_2_SO_4_ per well, and the optical density (OD) was determined at 405 nm using a microplate reader (Sunrise™, Tecan, Männedorf, Switzerland).

### 2.11. Quantitative Real-Time PCR (qRT-PCR) for Cytokines

Peripheral blood mononuclear cells (PBMCs) were isolated using the chicken PBMC isolation kit (P8910, Solarbio, Shanghai, China). Briefly, anticoagulant blood was mixed with sample diluent in a 1:1 volume ratio. The diluted blood samples were added to the liquid surface of the separation solution and centrifuged at 2500 rpm for 20–30 min. After centrifugation, the second layer of annular milky lymphocyte layer was carefully taken out with a pipette, washed with cell cleaning solution three times, and finally suspended in 0.5 mL of RPMI 1640 complete medium containing 10% FBS and 1% penicillin/streptomycin/amphotericin B solution (C810KA3468, Sangon Biotech, Shanghai, China). The cell suspension was adjusted to 1 × 10^7^ cells/mL, and 100 μL of cell suspension was plated in each well of a six-well cell culture plate. Cultured PBMCs were then stimulated with 10 µg/mL purified IBV protein (obtained by differential centrifugation of IBV) for 48 h at 37 °C, and total RNA was extracted from the stimulated PBMCs and reversely transcribed into cDNA. Changes in cytokine levels of IFN-γ, IL-6, and IL-12 were calculated using the 2^−∆∆Ct^ method (ref). Primers were synthesized by Tsingke Biotechnology Co., Ltd, Yangzhou, China. The primers used are shown in Table 2.

### 2.12. Viral Shedding Detection

The cloacal, trachea, and lacrimal gland swabs were individually diluted in 900 μL PBS, and 200 μL was inoculated into the allantoic fluid of 10-day-old chicken embryos. After 144 h, the allantoic fluid was collected and treated with 1% trypsinase at 37 °C for 3 h. Viral shedding was detected by the HA test. Clear and consistent HA was considered positive [32]. The samples with an HA titer of at least 4 HA units per 50 µL were interpreted as positive in this study. The viral shedding rate was calculated as the number of swabs that were found to be positive in the virus cloacal, trachea, or lacrimal gland divided by the number of all swabs taken from cloacal, trachea, or lacrimal gland, respectively.

### 2.13. Histopathology

The samples of the trachea, lung, and kidney were fixed with 10% formalin and dehydrated with escalating concentrations of ethanol solutions (70%, 80%, 90%, 95%, and 100%, respectively). Subsequently, the samples were embedded in paraffin and thinly sliced using a rotary microtome (RM2255, Leica, Berlin, Germany). After conventional HE staining, the samples were observed under an optical microscope. The pathological damage of different tissues was scored according to the following criteria. Trachea: 0: no lesions; 1: slight loss of tracheal epithelial cilia; 2: exfoliation of tracheal epithelial mucosa; 3: complete exfoliation of trachea epithelial cells. Lung: 0: no lesions; 1: small amount of hemorrhage in alveolar cavity and alveolar epithelial exfoliation; 2: extensive bleeding and inflammatory cell infiltration; 3: pulmonary interstitial hyperplasia, alveolar septum widening. Kidney: 0: no lesions; 1: bleeding and shedding of renal tubular epithelial cells; 2: inflammatory cell infiltration; 3: degenerative and necrotic renal tubules and collecting tubules [2,33].

### 2.14. Statistical Analysis

The statistical significance of differences between groups was determined using GraphPad Prism 8.0.2 (GraphPad Software, Inc., Boston, MA, USA). Comparative studies between multiple groups were performed using the one-way ANOVA test. *p* < 0.05 indicates a significant difference, while *p* < 0.01 indicates a highly significant difference.

## 3. Results

### 3.1. Construction and Characterization of Regulated Delayed Attenuation and Lysis S. typhimurium χ11246(pYA4545-S1)

To enhance the safety and effectiveness of the vaccine candidate, the *S. typhimurium* strain χ11246, with a regulated delayed attenuation and lysis system, was used. The system utilizes a tightly regulated *araC* P_BAD_ activator-promoter [26] for arabinose-dependent synthesis of AsdA and MurA enzymes [34]. The strain with the *asdA* mutation can grow by adding exogenous DAP, whereas *murA* is essential [34]. Hence, a conditional lethal arabinose-dependent *murA^−^* mutation is constructed by replacing the chromosomal *murA* promoter with the *araC* P_BAD_ activator-promoter [23]. The *asdA* gene was deleted in the chromosome, *murA* was arabinose-regulated, and additional mutations were introduced to enhance the complete lysis and antigen delivery. The Δ*asdA* mutation and Asd^+^ plasmid pYA4545 consist of a balanced lethal system. The *s1* gene of avian IBV was cloned into plasmid pYA4545 (Figure 1A) to generate recombinant plasmid pYA4545-S1 (Figure 1B). Double-enzyme digestion confirmed that the recombinant plasmid pYA4545-S1 was constructed successfully (Figure 1C). The Δ*sifA* mutation is a defined in-frame deletion of the *sifA* gene. It was introduced into strain χ11017 by phage P22 transduction from (χ8926::pYA3716) to generate strain χ11246 [23]. The presence of the mutation was verified by PCR (Figure 1D). Then, the Asd^+^ plasmids pYA4545-S1 and pYA4545 were transformed into strain χ11246 and maintained in LB broth with arabinose at 37 °C.

### 3.2. Expression of Recombinant S1 Protein and Collection of Antisera

To generate polyclonal antibodies for subsequent IFA test and Western blot analysis, we first identified a highly immunogenic region (909 bp) of IBV S1 protein using DNAstar 11.2.1 software (DNAstar Inc., Madison, WI, USA) based on the protein characteristics such as structural domains, hydrophilicity residues, and antigenicity. The region was cloned into the pET28a vector to generate plasmid pET28a-S1 and expressed in *E. coli* strain BL21 after induction with IPTG. Following induction, the bacteria were lysed using sonication, and the resulting lysate was centrifuged to separated supernatant and precipitation fractions. SDS-PAGE analysis was performed (Figure 2A), followed by Western blot (Figure 2B) using an anti-His tag antibody. The *E. coli* BL21 carrying the plasmids pET28a-GAPDH (~37 kDa) and pET28a were used as positive and negative control, respectively. An approximate 35 kDa band in the precipitate faction was observed, indicating high expression of the S1 protein as an inclusion body. To produce polyclonal antibodies, BALB/c mice were immunized with soluble S1 protein obtained after renaturation of the inclusion body. The mice sera were collected and tested for IBV-specific antibodies by ELISA, yielding a titer of 1:25,600. The result demonstrates that the high sensitivity of the antisera could be used as the primary antibody in subsequent experiments.

### 3.3. Expression of pYA4545-S1 In Vitro

An important factor affecting the immunity of the exogenetic antigen delivered by a DNA vaccine is the plasmid size since large plasmids may decrease the efficiency of plasmid translocation from the cytoplasm to the nucleus, where transcription occurs [35]. To verify the production of the S1 protein in Asd^+^ plasmid pYA4545-S1, both the IFA test and Western blot were performed using the anti-S1 antiserum developed in this study. As shown in Figure 3A,B, cells transfected with plasmid pYA4545-S1 exhibited strong fluorescence, while no fluorescence was observed when the cells were transfected with plasmid pYA4545. Furthermore, the Western blot results demonstrated the successful production of the S1 protein (~59 kDa) in cells after transfection with pYA4545-S1, while no band was detected after transfection with plasmid pYA4545. As a control, corresponding β-actin expression was also observed (~42 kDa) (Figure 3C). Based on the IFA and Western blot finding, the plasmid pYA4545-S1 can be synthesize S1 in HD11 cells.

### 3.4. Antibody Responses after Immunization with χ11246(pYA4545-S1)

The level of serum IgG antibodies reflects the extent of the humoral immune response induced by the vaccine [36]. The serum IgG antibody level in the χ11246(pYA4545-S1) immunized group was significantly higher than that in the χ11246(pYA4545) immunized group and the blank control group at 4 weeks after the first immunization (two weeks after the second immunization) (*p* < 0.01). The second immunization further increased serum IgG titers in the χ11246(pYA4545-S1) immunized group at 6 weeks after the first immunization (four weeks after the second immunization) (*p* < 0.01) (Figure 4A). These results indicated that SPF chickens immunized with χ11246(pYA4545-S1) could effectively generate serum IgG antibodies and induce humoral immunity.

### 3.5. Changes of Cytokine Genes Expression Pattern

Cytokines are crucial regulators of both innate and adaptive immune systems and can bind to specific cell surface receptors to initiate cascades of intracellular signaling for specific immune functions [37]. To study the effect of χ11246(pYA4545-S1) on induced immune-related cytokines in the host, isolated peripheral blood mononuclear cells from the experimental chickens were cultured with purified IBV protein for 48 h in six-well plates. Total RNA was extracted for qRT-PCR. The expression levels of Th1-type cytokines IL-12 (*p* < 0.05), IFN-γ (*p* < 0.05), and Th2-type cytokines IL-6 (*p* < 0.01) in the χ11246(pYA4545-S1) immunized group were significantly higher than those in the χ11246(pYA4545) immunized group (Figure 4B). This indicates engagement of both Th1 (cellular) and Th2 (humoral) immune responses upon χ11246(pYA4545-S1) immunization.

### 3.6. Orally Administered χ11246(pYA4545-S1) Provides Protection against 793B IBV Challenge

Protection assays were conducted to determine whether the *S. typhimurium* χ11246(pYA4545-S1) vaccine candidate protects SPF chickens against IBV infection. Compared with the virus-challenged group, the viral shedding ratios in the lacrimal gland (*p* < 0.01), trachea (*p* < 0.05), and cloaca (*p* < 0.05) of the χ11246(pYA4545-S1) immunized group were significantly lower than those in the χ11246(pYA4545) immunized group and the virus-challenged control group (Figure 5). Compared to the number of affected chickens in the χ11246(pYA4545) immunized group and the virus-challenged group, chickens immunized with χ11246(pYA4545-S1) exhibited protection ratios of 72%, 56%, and 56% in the lacrimal gland, oropharynx, and cloaca, respectively (Table 3). Strain χ11246(pYA4545) also conferred a low level of protective ratio, possibly due to the immune enhancement resulting from immunity to the *S. typhimurium* vector. Histological analysis revealed no lesions in any of the organs collected from the χ11246(pYA4545-S1) immunized group. In contrast, chickens in the χ11246(pYA4545) immunized group and the virus-challenged control group exhibited obvious lesions in all collected tissues. The trachea displayed extensive exfoliation of ciliated columnar epithelium (Figure 6A). In the lungs, interstitial hyperemia and alveolar epithelium abscission were observed in the χ11246(pYA4545) immunized group and the virus-challenged control group (Figure 6B). In the kidneys, the most prominent histopathological changes were nuclear concentration and degeneration or necrosis of renal tubules (Figure 6C). In contrast to the χ11246(pYA4545) immunized group and the virus-challenged control groups, the tissues in the χ11246(pYA4545-S1) immunized group exhibited histologically similarities to normal tissues. Histological scores showed that the lesion scores of the χ11246(pYA4545-S1) immunized group were significantly lower than those of the virus-challenged control group and the χ11246(pYA4545) immunized group (*p* < 0.01) (Figure 7). Collectively, these results demonstrate that immunization with χ11246(pYA4545-S1) provides significant protection against IBV challenge in the SPF chicken model, suggesting its potential application in vaccination strategies against IBV infection.

## 4. Discussion

The development of an effective vaccine for the control of IBV is of great importance to the poultry industry worldwide [38]. Zhang et al. [39] studied the classical vaccines serial passage to 105 generations, which can provide effective protection. In addition, many DNA vaccines against IBV have been studied, using S1 [4,38,40], M [41,42], or N [43] protein as protective antigens, and demonstrate varying levels of protection against IBV. Lei et al. [44] assessed the effect of cross-immunization with different serotype commercial vaccines and different immunization procedures, indicating that it is critical to develop the appropriate matching vaccine strains and determine the most appropriate vaccination regime. Buharideen et al. [45] studied the different vaccination strategies on vaccine efficacy; they demonstrated that four doses of the attenuated vaccine and booster immunization with an inactivated vaccine resulted in better immunity. In this study, similar immune effects can be achieved by oral immunization three times. Notably, the plasmid pYA4545 has a nuclear-targeting sequence, SV40 enhancer, strong cytomegalovirus (CMV) immediate-early gene enhancer/promoter (CMV E/P), and SV40 poly A sequence, which can prevent nuclease degradation and facilitate entry into the nucleus, resulting in high-level sustained protein production, and ultimately inducing protective immunity against pathogens [46]. Moreover, in contrast to intramuscular injections of DNA vaccines, *S. typhimurium* can be administrated orally, enabling convenient immunization through mixed feeding or mixed drinking in poultry farms. Previous studies have shown that this delayed attenuation and lysis system delivering influenza HA and NP antigens successfully induced complete protection against influenza virus challenge in mice [22,47]. In this context, this present study provides the first evidence of the efficacy of a vaccination strategy using regulated delayed attenuation and lysis *S. typhimurium* χ11246 to deliver the S1 protein of avian IBV, conferring protection against chicken infectious bronchitis.

*S. typhimurium* χ11246 contains a *sifA^−^* mutant, which helps it escape the endosome after invasion. The *sifA* gene is a *Salmonella* pathogenicity island 2 (SPI-2) encoded T3SS secretion effector that governs conversion of the SCV into filaments. The *sifA^−^* mutants lose the integrity of the SCV, allowing *Salmonella* to be released into the cytosol [48]. The regulated delayed attenuation and lysis strain χ11246 exits the endosome soon after invading a host cell and multiplies in the cytoplasm. Then, due to of the lack of MurA synthesis, it lyses to release the plasmid pYA4545-S1 into the cytosol. The cells then produce the S1 antigen encoded on the plasmid to induce immunity [22].

Humoral immune responses can inhibit viral replication and have been shown to correlate with IBV-specific antibody titers [36]. In this study, the serum IgG antibody level in the χ11246(pYA4545-S1) immunized group was significantly higher than that in the χ11246(pYA4545) immunized group and the blank control group, showing an upward trend over time (Figure 4A). IgG antibodies are important determinants of effective viral clearance [7]. The high IgG antibody titers contribute to the protective effect induced by the strain χ11246(pYA4545-S1) against IBV infection. These results indicate that vaccine candidate strain induces high systemic humoral immune responses.

The cytokines produced after antigenic stimulation in vitro are important parameters for defining the type of elicited immunity [14,49]. Our data showed that χ11246(pYA4545-S1) induced efficient IBV-specific cytokine responses involving both Th1 (IFN-γ and IL-12) and Th2 (IL-6), indicating a mixed type of immunity (Figure 4B). Th1 cells guide cell-mediated immune response, while Th2 cells guide the differentiation of B cells into plasma cells for antibody-mediated humoral immunity [50]. Among these cytokines, the Th1 cytokine IFN-γ exerts a potent antiviral effect and promotes the activation of natural killer (NK) cells and macrophages to inhibit IBV replication and transmission with the host [51]. Additionally, IFN-γ can upregulate MHC-I and MHC-II molecules and induce macrophages to produce IL-12, nitric oxide, and superoxide, all of which are closely related to pathogen clearance [14]. IL-12 plays a crucial role in driving the differentiation of naive T cells toward the Th1 direction, which can stimulate the proliferation of T cells and promote the differentiation of Th0 cells into Th1 cells to trigger the cytotoxic activity of CTL and NK cells. Furthermore, IL-12 promotes the secretion of IFN-γ, TNF-α, GM-CSF, and other cytokines critical for virus elimination [52]. The Th2 cytokine IL-6 is a pro-inflammatory cytokine whose production accelerates the inflammatory response period and enhances viral clearance [37,53]. The present study demonstrated that chickens vaccinated with χ11246(pYA4545-S1) exhibited higher levels of IL-6, IL-12, and IFN-γ compared to the control chickens. Overall, these results suggest that recombinant χ11246(pYA4545-S1) could induce potent cell-mediated immune responses.

Oral administration of recombinant χ11246(pYA4545-S1) strains can effectively activate both cellular and humoral immunities simultaneously. In the challenge test, *c*hickens vaccinated with χ11246(pYA4545-S1) exhibited a higher protection ratio and experienced less histopathological injury compared to the control groups. Assessing the impact of IBV infection on chickens involves examining viral shedding and pathological changes in various tissues [14,33,38], both crucial indexes for evaluating the efficacy of IBV vaccines. In this study, viruses were shed from the lacrimal gland, trachea, and cloaca in both the empty vector control and the virus-challenged control groups, with a higher ratio detected compared to the χ11246(pYA4545-S1) immunized group (Figure 5). Oral immunization with recombinant χ11246(pYA4545-S1) resulted in a significant protection percentage of 72%, 56%, and 56% protection in the lacrimal gland, trachea, and cloaca, respectively (Table 3). These findings are consistent with the previous studies conducted on chickens [9,14]. The protection might be attributed to the upregulation of humoral immunity and cytokines, as reflected in the aforementioned results, which contribute to inhibiting virus replication and promoting virus elimination. The efficacy of protection was also evident at the histological level. The tissue tropism of IBV is determined by the S protein, with different strains capable of replicating in the Harderian glands, upper respiratory tract, and kidney [54]. Our results revealed the presence of exfoliation in trachea epithelial cells (Figure 6A), congestion and lymphocyte infiltration in the lungs (Figure 6B), and degeneration and necrosis in renal epithelial cells (Figure 6C) in the virus-challenged control group and the χ11246(pYA4545) immunized group but not in the trachea, lung, and kidney of the χ11246(pYA4545-S1) immunized group. Additionally, the histopathological score of the χ11246(pYA4545-S1) immunized group was significantly lower than that of virus-challenged control group and the χ11246(pYA4545) group (Figure 7). The presence of systemic circulating antibodies is important in protecting against damage to epithelial tissue [55], which likely contributes to the safeguarding the integrity of tracheal and renal tubular epithelial cells in the χ11246(pYA4545-S1) immunized group. Overall, these results demonstrate that immunizing three times with the recombinant vaccine candidate strain effectively confers some degree of protection against IBV challenge.

## 5. Conclusions and Limitations

In conclusion, the recombinant *S. typhimurium* χ11246(pYA4545-S1) vaccine candidate strain, featuring the arabinose-regulated *murA* lysis system and the robust immunogenicity of the S1 protein, is capable of activating both cellular and humoral responses in the host, thereby providing protection against IBV infection. Accordingly, *S. typhimurium* χ11246(pYA4545-S1) provides a new strategy for the prevention of IB. Significantly, as a delivery vector of foreign antigens, an important issue to be considered is the safety of the regulated delayed attenuation and lysis *S. typhimurium*. This is the limitation of this paper. Therefore, it is suggested that the survival time of *S. typhimurium* χ11246 in vivo and its impact on the host be examined in future studies.

## Figures and Tables

**Figure 1 biomolecules-14-00133-f001:**
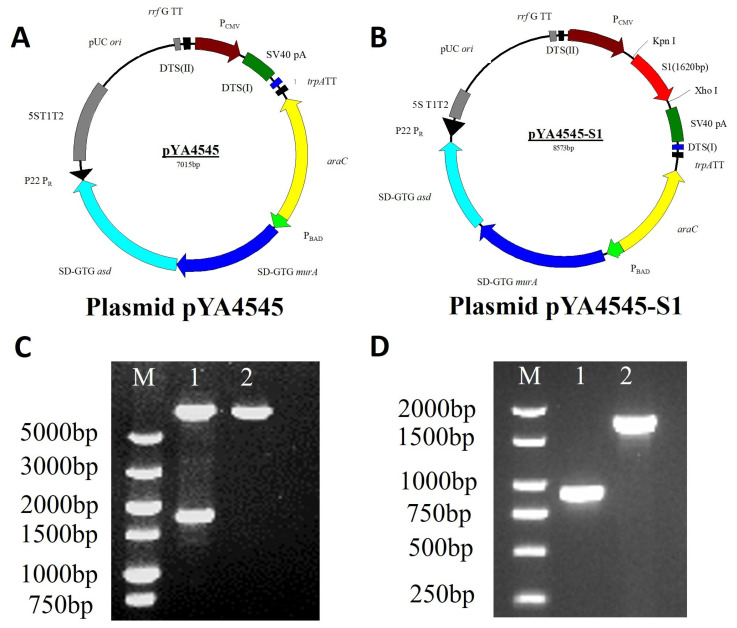
Plasmid maps and identification of χ11246 (pYA4545-S1). (**A**) Plasmid pYA4545. (**B**) Recombinant plasmid pYA4545-S1. (**C**) Identification of recombinant plasmid pYA4545-S1 by double-enzyme digestion (KpnI and XhoI). M: DL5000 DNA marker; Lane 1: pYA4545-S1 (1620 bp and 6953 bp); Lane 2: pYA4545. (**D**) The identification of delayed attenuation and lysis *Salmonella* χ11246. M: DL2000 DNA marker; Lane 1: The Δ*sifA* gene of χ11246 (850 bp); Lane 2: The *sifA* gene of UK-1 wild-type χ3761 (1683 bp). Original figures can be found in Appendix A.

**Figure 2 biomolecules-14-00133-f002:**
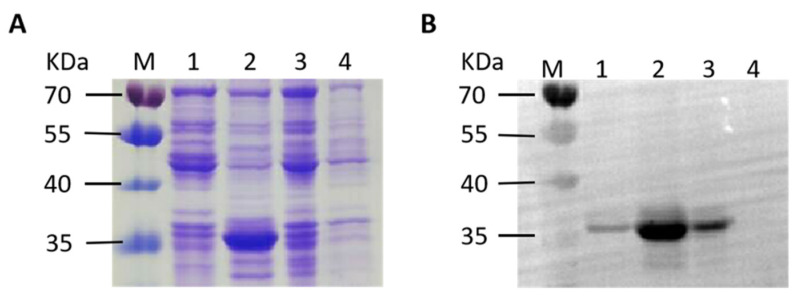
Expression of recombinant S1 protein. (**A**) SDS-PAGE analysis. (**B**) Western blot analysis. M: 180 kDa prestained protein marker; Lane 1: the supernatant of BL21(pET28a-S1); Lane 2: the precipitate of BL21(pET28a-S1); Lane 3: positive control BL21(pET28a-GAPDH); Lane 4: negative control BL21(pET28a). Original figures can be found in Appendix A.

**Figure 3 biomolecules-14-00133-f003:**
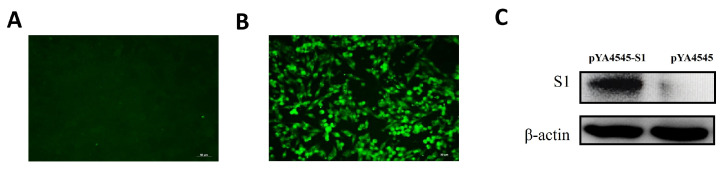
Indirect immunofluorescence and Western blot detection of the expressed S1 protein in HD11 cells. Transient expression protein was detected 48 h after transfection with the polyclonal antibody serum prepared in this study. Cells transfected with pYA4545-S1 (**B**) showed green fluorescence, while cells transfected with control vector pYA4545 (**A**) did not show fluorescence. Western blot (**C**) showed cells transfected with pYA4545-S1 had a specific band, while cells transfected with backbone plasmid pYA4545 did not. Original figures can be found in Appendix A.

**Figure 4 biomolecules-14-00133-f004:**
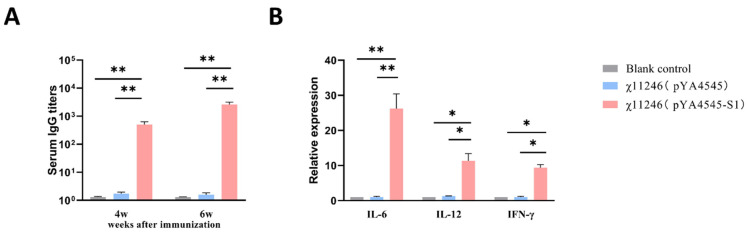
(**A**) Serum IgG antibody was detected by indirect ELISA. (**B**) Expression of IL-6, IL-12, and IFN-γ genes in peripheral blood mononuclear cells was detected by qRT-PCR. Significant differences are indicated (* *p* < 0.05; ** *p* < 0.01). Data are expressed as the mean ± standard deviations of the six chickens in each group.

**Figure 5 biomolecules-14-00133-f005:**
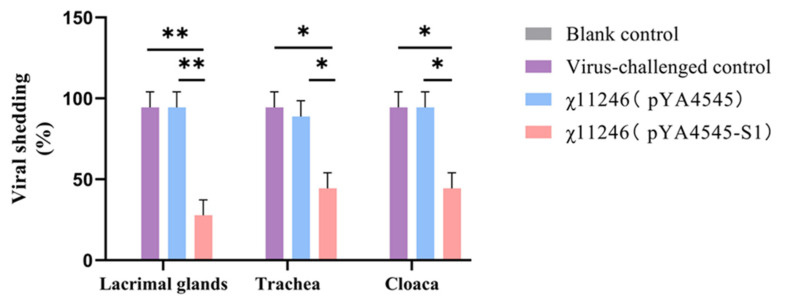
Viral shedding ratio of lacrimal gland, trachea, and cloaca in each group. Significant differences are indicated (* *p* < 0.05; ** *p* < 0.01). Data are expressed as the mean ± standard deviations of the six chickens in each group. Note: viral shedding = the average number of affected chickens × 100%.

**Figure 6 biomolecules-14-00133-f006:**
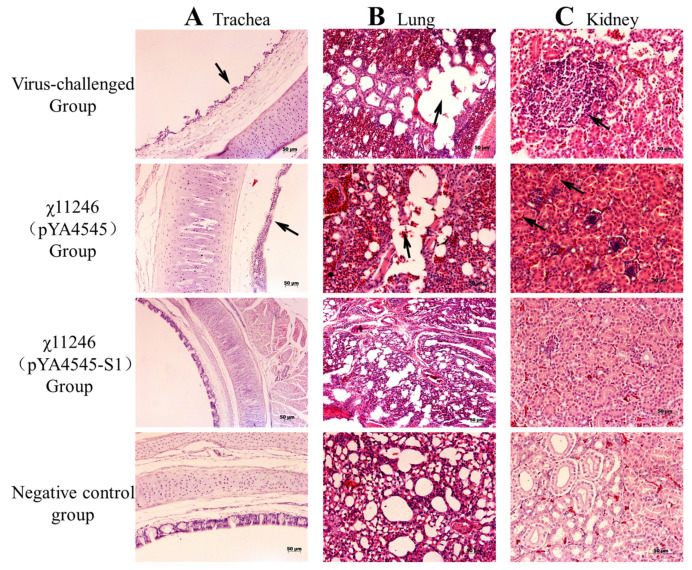
Photomicrographs of hematoxylin-and eosin-stained trachea, lung, and kidney sections of chickens on 5th day post-IBV challenge. Chickens (*n* = 6/per group) were immunized with sterile PBS, χ11246(pYA4545-S1), and χ11246(pYA4545), and 6 weeks later, all the vaccinated chickens were challenged with 10^4^ EID_50_ IBV. At 5th day post-challenge, chickens (*n* = 6/per group) were sacrificed, and the trachea, lungs, and kidneys were collected for histological analysis. Black arrows represent lesions.

**Figure 7 biomolecules-14-00133-f007:**
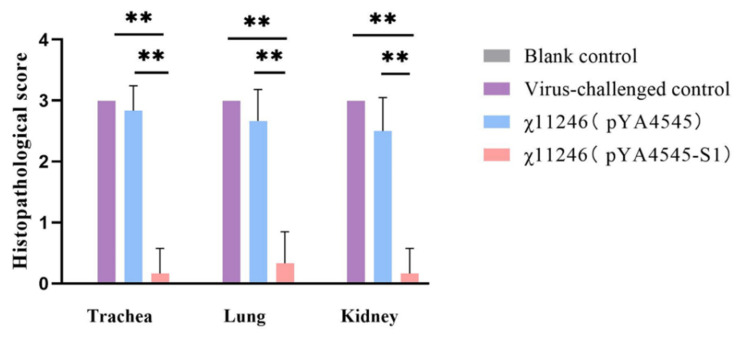
Histopathological score of the different groups. Significant differences are indicated (** *p* < 0.01). Data are expressed as the mean ± standard deviations of the six chickens in each group.

**Table 1 biomolecules-14-00133-t001:** Bacterial strains and plasmids used in this study.

Strains, Plasmid, or Primer	Characteristics	Sources or References
*E. coli.* strains		
BL21 (DE3)	For expression of the recombinant plasmids	Invitrogen
χ7213	*thi 1 thr 1 leuB6 fhuA21 lacY1 glnV44 asdA4 recA1* RP4 2 Tc:: Mu[λ pir]Δ*asdA4* (Δ*zhf-2::Tn10*)	[22]
*S. enterica* serovar *Typhimurium* UK-1 strains		
χ11246	Δ*asdA*::TT *araC* P_BAD_*c2* Δ*araBAD* Δ*(gmd-fcl)* Δ*pmi* Δ*relA*::*araC* P_BAD_*lacI*, TT ΔP_murA_::TT *araC* P_BAD_*murA* Δ*sifA*	[23]
Plasmids		
pYA4545	pUC *ori araC* P_BAD_ SD-GTG *murA* SD-GTG *asdA* P22 P_R_ antisense mRNA eukaryotic expression cassette of with DTS (I), DTS (II), and SV 40 polyA	[23]
pET28a	Expression vector	Novage
PMD-19T	Cloning vector; Amp^r^	TaKaRa
pET28a-S1	A recombinant expression vector containing S1; Kan^r^	This study
pYA4545-S1	pYA4545 with *s1* gene	This study

**Table 2 biomolecules-14-00133-t002:** Primers used in this study.

Primers	Sequences (5′–3′)	Product Size (bp)	Function
S1-P1	CGGGGTACCATGTTGGGCAAACCGCTTTTACT	1620	Cloning of plasmid pYA4545-S1
S1-P2	TAAAATAACCTCTTGTGCGGTTCCATTAATAAAGTAGGCTAGGGCTT		
S1-P3	AAGCCCTAGCCTACTTTATTAATGGAACCGCACAAGAGGTTATTTTA		
S1-P4	CCGCTCGAGAGAACGTCTAGAGCGACGTGTTCCGTT		
S1-F1	ATGGGTCGCGGATCCGAATTCTTTACTACTACCAAAGTGCCTTTAGGC	909	Cloning of plasmid pET28a-S1
S1-R1	GTGGTGGTGGTGGTGCTCGAGTGGGTGGTATGACCCATACATAAA		
ΔsifA-F1	TGATGAGCTCTTTCTCTTCTCCAAAATCTC	850	
ΔsifA-R1	CTTAGGTACCGGTCGATTTAATCAATTATG		
sifA-F	CTCTTCTCCAAAATCTCTCCAAA	1683	
sifA-R	CATTACGCTGACCATTGTGA		
β-Actin	F: TATGTGCAAGGCCGGTTTC R: TGTCTTTCTGGCCCATACCAA	110	
IFN-γ	F: TGAGCCAGATTGTTTCGATG R: CTTGGCCAGGTCCATGATA	152	
IL-6	F: CAAGAAGTTCACCGTGTGCGAGA R: ATTCCAGGTAGGTCTGAAAGGCG	254	
IL-12	F: CGAAGTGAAGGAGTTCCCAGAT R: GACCGTATCATTTGCCCATTG	123	

Underlined nucleotides denote enzyme restriction sites.

**Table 3 biomolecules-14-00133-t003:** Protection ratios for each group challenged by the 793 serotype IBV.

Groups	Sample Type	No. of Affected ^a^			Protection Rate (%) ^c^
		1 ^b^	2 ^b^	3 ^b^	
Blank control	Lacrimal gland	0/6	0/6	0/6	/
	Trachea	0/6	0/6	0/6	
	Cloaca	0/6	0/6	0/6	
Virus-challenged control	Lacrimal gland	6/6	5/6	6/6	/
	Trachea	6/6	6/6	5/6	
	Cloaca	6/6	6/6	5/6	
χ11246(pYA4545)	Lacrimal gland	6/6	6/6	5/6	6%
	Trachea	6/6	5/6	5/6	11%
	Cloaca	5/6	6/6	6/6	6%
χ11246(pYA4545-S1)	Lacrimal gland	2/6	1/6	2/6	72%
	Trachea	3/6	2/6	3/6	56%
	Cloaca	3/6	3/6	2/6	56%

^a^ Affected chickens with virus identified by HA test. ^b^ The vaccine protection assays were repeated three times (six chickens per group). ^c^ Protection ratio was determined by the average of the number of chickens without viral shedding/total number of chickens.

## Data Availability

All data generated or analyzed during this study are included in this article.

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
