# Peer review of "Salmonella typhimurium Vaccine Candidate Delivering Infectious Bronchitis Virus S1 Protein to Induce Protection"

_biomolecules, 2024, doi:10.3390/biom14010133_

Round 1
Reviewer 1 Report
Comments and Suggestions for Authors
Very interesting and very good work on the new generation of vaccines based on biotechnology technics.the results obtained are very promising for a better control of the most wide spread virus in chickens over the last eight decades. However it was pity the authors missed: (1) to evaluate respiratory symptoms during the 5 days post challenge; (2) to evaluate viral detection and viral load in in trachea and cloacal swabs at 5 days post challenge using qRT-PCR; (3) to evaluate cilia activity of trachea at 5 days post challenge. these criteria are mostly used while evaluating IBV vaccines efficacy.
Discussion: further research needed; (1)to evaluate this recombinant vaccine in layers and to evaluate its protection on egg production and egg quality post challenge; (2) comparison with classical vaccines ( live attenuated and inactivated)
Comments on the manuscript
Ligne 32: IBV does not cause disease of digestive tract or the bursa of Fabricius ( to be omitted)
ligne 205 and 243: how many eggs ( embryos) were inoculated for each dilution; ligne 243 :swabs were treated individually or pooled.
ligne 388: χ11246(pYA4545) saoul be replaced by χ11246(pYA4545-S1)
ligne 412: Important should replaced by Importance
Ligne 497: "Confer protection" should replaced by " confer some degree of protection" or some level of protection.
Reviewer 2 Report
Comments and Suggestions for Authors
Vaccination is an important tool to prevent poultry infectious diseases, while the currently available vaccines used to protect against infectious bronchitis are considered efficient. The literature contains numerous data to support the efficacy of the available vaccines. As the cross-protection is limited, the virus serotype is very important when selecting the vaccinal strain, homologous strain vaccines inducing higher protection
Ali, A.; Hassan, M.S.H.; Najimudeen, S.M.; Farooq, M.; Shany, S.; El-Safty, M.M.; Shalaby, A.A.; Abdul-Careem, M.F. Efficacy of Two Vaccination Strategies against Infectious Bronchitis in Laying Hens. Vaccines 2023, 11, 338. https://doi.org/10.3390/vaccines11020338
Zhang, X., Chen, T., Chen, S. et al. The Efficacy of a Live Attenuated TW I-Type Infectious Bronchitis Virus Vaccine Candidate. Virol. Sin. 36, 1431–1442 (2021). https://doi.org/10.1007/s12250-021-00419-2
Shao, L., Zhao, J., Li, L. et al. Pathogenic characteristics of a QX-like infectious bronchitis virus strain SD in chickens exposed at different ages and protective efficacy of combining live homologous and heterologous vaccination. Vet Res 51, 86 (2020). https://doi.org/10.1186/s13567-020-00811-y
From this point of view, the justification of this study is weak, the authors should modify the text and address the relevance of their research
Furthermore, several references used to justify the research are dated.
The authors should add the euthanasia protocol
This study includes a complex protocol for efficacy evaluation, but no data on the safety
The authors should add the limitations of the study.
The conclusion " .... lays the foundation for future development of an IB vaccine" is an overstatement, it should be rephrased
Overall, the manuscript does not present the scientific level of the current journal
Author Response
Please see the attachment, thank you.

Reviewer 3 Report
Comments and Suggestions for Authors
The manuscript is generally well written and organised. My only comment is about the lack of discussion about why the χ11246(pYA4545) group also showed low but levels of protection. I suggest a brief comment on this.
- How is the histophatological score assessed? I did not see the explanation of this. Please, mention this or refer to previous publications.
Comments on the Quality of English LanguageThe manuscript is generally well written and organised. My only comment is about the lack of discussion about why the χ11246(pYA4545) group also showed low but levels of protection. I suggest a brief comment on this.
- How is the histophatological score assessed? I did not see the explanation of this. Please, mention this or refer to previous publications.
Author Response
Please see the attachment, thank you.

Reviewer 4 Report
Comments and Suggestions for Authors
COMMENTS ARE INCLUDED IN THE MANUSCRIPT

Author Response
Please see the attachment, thank you.

Round 2
Reviewer 2 Report
Comments and Suggestions for Authors
The authors performed certain modifications, but flaws are still found in the revised form of the manuscript.
The overall approach of presenting the lack of efficacy and safety of the currently available vaccines is not supported by the results of numerous studies and field results.
Regarding the safety, the authors have little justification as their product has no safety assessment (comment 3)
Comment 1 was intended to suggest to the authors that they should rephrase all the manuscript sections, phrases and words, the literature was provided to a better understanding of the subject
The authors replied: Response 1: Thank you for your careful review. We have studied the comments carefully and updated the reference. The revision was marked in red. (line32-33, line 45-50, line 416-422).
Unfortunately, the manuscript still includes phrases and paragraphs that declare that the currently available vaccines are not efficient, and this statement is not accurate.
If the authors want to justify their study, this approach is not ethical, and not accurate for the particular case of IB vaccination.
Eg lines 13-15 “ABSTRACT: Currently, the main vaccines used to control avian infectious bronchitis (IB) are live attenuated and inactivated vaccines. However, their clinical application is limited due to the safety concerns and the poor immunogenicity. An alternative vaccine against IB is urgently needed to pro-…”
Several commercial companies produced efficient vaccines.
Lines 47-48 “On the other hand, inactivated IBV vaccines require multiple immunizations to maintain immunity [13], often necessitating co-administration with live attenuated vaccines and high-dose adjuvants [14].”
This statement is not accurate either, the authors should upgrade their level on IBV vaccines and the immune response particularities in chicken
Furthermore, what is the scientific value of such phrases:
“Currently, the main vaccines used to control avian infectious bronchitis (IB) are live attenuated and inactivated vaccines.”
“Together, this study provides a new idea for future development of an IB vaccine. “
“Further research is necessary to evaluate the effect of S. Typhimurium 510 χ11246(pYA4545-S1) vaccine candidate on egg production and quality in layers.”
“… confer some degree of protection against IBV challenge. “
The current vaccines are not perfect, but their minuses and disadvantages are not as the authors present them. Efficacy or induced protection against IBV infection is generally presented as dependent on the administration of the homologous vaccine.
Repeated boosters are in general required in poultry, this aspect is not characteristic or specific for IB vaccination, but for most vaccines licensed in poultry (this is connected to the immune system particularities)
The authors added and marked with red references, but they preserved the dated ones.
Comments 2: The authors should add the euthanasia protocol.
The authors added a line that mentions euthanasia, but the protocol is not described
This is not sufficient for any ISI journal level with ethical standards for studies involving animals.
Comments 3: This study includes a complex protocol for efficacy evaluation, but no data on the safety.
Response 3: Thank you for your careful review. In the following studies, we will add the experiments to detect sustained survival time of S. Typhimurium in immunized chickens after immunization and the detection of indicators of the impact of S. Typhimurium shedding on external environmental.
The response does not improve the scientific value of the manuscript.
Comments 4: The authors should add the limitations of the study.
Response 4: Thank you for your comments. In this study, we lack of the comparison with classical vaccines (live attenuated and inactivated). We will make up for this in future studies. But we discuss the advantages of this recombinant vaccine over other classical vaccines in the Introduction based on other references (line 42-49) to make up for it as far as possible. Limitations of this study include the evaluation of tissue viral load and cilia activity after challenge, we will overcome these limitations in our future studies. Nevertheless, We still believe this study can provide a reference for IB vaccine development
The response does not improve the scientific value of the manuscript. The study should include limitations.
New comment for the text – lines 259-266 – the authors should add the references for the criteria used to obtain lesion scoring.
Author Response
Please see the attachment, thank you.
